# Revolutionizing EMCCD Denoising through a Novel Physics-Based Learning Framework for Noise Modeling

**Haiyang Jiang[1], Tetsuichi Wazawa [2], Imari Sato [3], Takeharu Nagai [2], Yinqiang Zheng [1]** [†]
[1]The Univerisity of Tokyo
[2]Osaka University
[3]National Institute of Informatics

## Abstract

Electron-multiplying charge-coupled device (EMCCD) has been instrumental in sensitive observations under low-light situations including astronomy, material science, and biology. Despite its ingenious designs to enhance target signals overcoming read-out circuit noises, produced images are not completely noise free, which could still cast a cloud on desired experiment outcomes, especially in fluorescence microscopy. Existing studies on EMCCD's noise model have been focusing on statistical characteristics in theory, yet unable to incorporate latest advancements in the field of computational photography, where physics-based noise models are utilized to guide deep learning processes, creating adaptive denoising algorithms for ordinary image sensors. Still, those models are not directly applicable to EMCCD. In this paper, we intend to pioneer EMCCD denoising by introducing a systematic study on physics-based noise model calibration procedures for an EMCCD camera, accurately estimating statistical features of observable noise components in experiments, which are then utilized to generate substantial amount of authentic training samples for one of the most recent neural networks. A first real-world test image dataset for EMCCD is captured, containing both images of ordinary daily scenes and those of microscopic contents. Benchmarking upon the testset and authentic microscopic images, we demonstrate distinct advantages of our model against previous methods for EMCCD and physics-based noise modeling, forging a promising new path for EMCCD denoising[1].

## 1 Introduction

Owing to its high sensitivity to photons and effective amplification of signals by impact ionization during charge transfer at high frame rates, EMCCD has, since its invention by Denvir & Conroy (2003), become increasingly significant to multiple research areas that require experiment observations under extreme low-light conditions, including astronomy (Mallik et al., 2019; Tubbs, 2003), quantum imaging (Jeong et al., 2023; Meda et al., 2017; Kumar et al., 2017), and fluorescence microscopy in biology (Chao et al., 2016; Lee et al., 2017; Han et al., 2012; Kagan et al., 2022; Wang et al., 2023; Yokota et al., 2021).Yet, electron multiplication registers inside an EMCCD do not guarantee immunity of noise in resulting images. As can be seen from figure 1, although in sub-image (a) the 16-bit raw image doesn't reveal much information [2], noise patterns can be observed clearly when contrast being enhanced in (b). Such noise pattern cannot be successfully removed through averaging operation, as shown in (c). Exposing for a longer period, or including more frames into averaging, is not necessarily a viable solution as well, especially in the field of fluorescence microscopy, since subjects being observed might be susceptible to damage from fluorophores as pointed out by Icha et al. (2017); Hopt & Neher (2001), while molecular level motions can cause

---

[†]Corresponding author: `yqzheng@ai.u-tokyo.ac.jp`.

[1]Code and dataset can be found at this page
[2]Pixel values are typically controlled to be within a magnitude of thousands in order to protect a device from EM gain aging, much lower than the maximum value of 16-bit integers, *i.e.*, 65535.

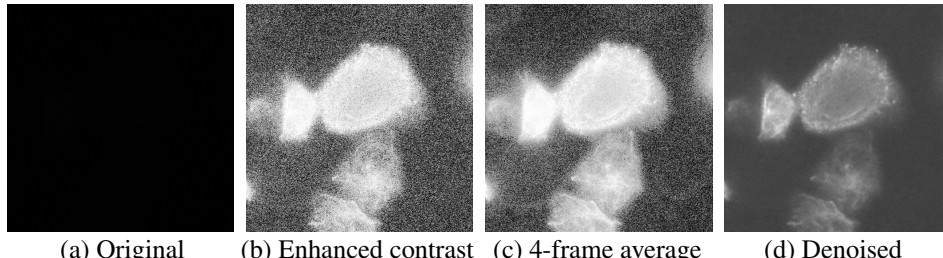

(a) Original     (b) Enhanced contrast    (c) 4-frame average     (d) Denoised

Figure 1: Our denoising result. (a) Original 16-bit raw image; (b) Contrast enhanced version to show noise patterns; (c) Averaged result over 4 consecutive frames with residual noise; (d) Denoised clean output of our system.

misalignment across long image sequences. Thus, a reliable denoising algorithm for EMCCD would be beneficial to scientific community trying to harness full power of the device.

Researchers and manufacturers have been studying detailed mathematical models of EMCCD over the years, encompassing accurate gain calibration (Ryan et al., 2021; Qiao et al., 2021), noise components (Basden, 2015; Hirsch et al., 2013; Dussault & Hoess, 2004), and signal-to-noise ratio (SNR) analysis (Zhang & Chen, 2009; Plakhotnik et al., 2006; Basden et al., 2003), *etc*. With the advancements of deep learning techniques in recent years, researches involving EMCCD have embraced this trend, developing task-specific models to better serve distinct objectives in experiments. A common denoising model or paradigm, however, has yet to be put forward for EMCCDs.

On the other hand, researches in the field of computer vision have progressed to simulate adequately precise noise patterns for standard optical camera sensors from a physics-based perspective (Wei et al., 2021; Feng et al., 2024; Cao et al., 2023; Ji et al., 2024; 2023b;a), in order to synthesize authentic training samples for deep neural networks that can achieve comparable or better performances than those trained with real-world dataset, like **S**ee-**i**n-the-**D**ark (**SID**) by Chen et al. (2018). Nonetheless, those models are intrinsically different from physics in EMCCDs.

In this paper, we intend to derive insights from the progress of noise modeling in computer vision in order to boost image quality of EMCCD cameras fundamentally. We propose a generic and versatile noise model for EMCCD, incorporating theoretical analysis on its physical composition and direct experiment results during calibration process on an iXon Ultra 897 EMCCD manufactured by Andor, whose products have been widely adopted in both academic and industrial research. A systematic approach for noise parameter estimation is also detailed, facilitating applicability of our pipeline onto devices from various manufacturers. With acquired statistical features of the EMCCD noise pattern, we dynamically generate massive amount of noisy-clean image pairs from ground truth of a low-light raw image dataset, SID, upon which one of the latest powerful transformer model, Uformer from Wang et al. (2022), is trained to extract desirable signal from noises with dexterity. Because SID consists only of scenes on a macroscopic scale, we further finetuned our model using synthetic microscopic data, proving its flexibility of being transferred from macro world to fluorescence microscopy. For a fair comparison to existing methods, we utilize our EMCCD to shoot 224 noisy-clean image pairs using long-short exposure method similar to SID. This world's first EMCCD testset contains 15 exposure ratios [3], ranging from 30 to 1600. Quantitative evaluations on it have demonstrated advantages that our method possesses over both physics-based noise modeling in computer vision that cannot be properly generalized to EMCCD, and theoretical noise analysis for EMCCD that cannot pinpoint valuable characteristics for deep learning models.

Our contributions are three-fold:

1. A novel noise model for generating authentic noisy images specific to EMCCD is proposed, integrating fundamental analysis on structural design of the camera, as well as pixel level behaviors from images produced by EMCCD during experiments.

2. We establish a unique pipeline from noise parameter calibration to training strategy of utilizing state-of-the-art enhancement networks on synthetic dataset, connecting physics perspectives and algorithmic designs for EMCCD denoising in fluorescence microscopy.

---

[3]Quotient of long exposure time for ground truth divided by short exposure time for noisy image, indicating differences in light intensities, *i.e.*, magnitude of photon counts, as well as difficulties for denoising.

    3. Experiments are conducted, testing our model against previous methods on a collected first real-world dataset for EMCCD with macroscopic and microscopic scenes, demonstrating superiority of our noise model and training pipeline.

# 2 RELATED WORK

## 2.1 EMCCD NOISE

As a relatively new imaging system, EMCCD has been studied with advanced mathematical and signal processing tools since its birth by Denvir & Conroy (2003). One unique feature of EMCCD noise is multiplicative noise induced by electron multiplication registers (Basden et al., 2003), which can be modeled by a Poisson-gamma distribution (Hirsch et al., 2013). This component can be described by a noise factor (Denvir & Conroy, 2003), amplifying Poisson distributed shot noise caused by quantum essence of photons to corresponding level. Theoretical analysis by Zhang & Chen (2009) gives that noise factor of EMCCD is $\sqrt{2}$, while in practice Dussault & Hoess (2004); Araújo et al. (2018) observed and analyzed that existence of register voltage degradation can bring such value down to $1.3$. Zhang & Chen (2009) characterized 5 noise sources in EMCCD: photon shot noise, multiplicative noise, dark current noise, clock induced charge, and readout noise. Hirsch et al. (2013) delved deeper into physical structures of noise sources, presenting a Poisson-gamma-normal (PGN) distribution model, for mathematical representation of each noise element. Ryan et al. (2021) established a gain series method based on the EMCCD noise model for more accurate EMCCD calibration than mean-variance intensity series method.

Another artifact in EMCCD images worth mentioning, yet not commonly discussed, is that, inheriting from CCD sensors' blooming effect, EMCCD produces vertical/horizontal light streaks in high-contrast situations (Mallik et al., 2019). This phenomenon is believed to be caused by the absence of a mechanical shutter. Thus, we include algorithmic anti-blooming as a part of our denoising process, to account for all possible discrepancies between noisy and clean images.

## 2.2 PHYSICS-BASED IMAGE SENSOR NOISE MODEL

Noise in ordinary camera image sensors under extreme low-light can be divided into signal dependent and signal independent components (Nakamura, 2005). Recent noise models draw correlation between noise components and camera gain. **E**xtreme **L**ow-light **D**enoising (**ELD**), by Wei et al. (2021), identified 4 elements for noises in raw sensor data: shot noise, row noise, read noise, and quantization noise. PMN, by Feng et al. (2024), further proposed the existence of dark shading in digital image sensors, depicting fixed-pattern noise (FPN) on fine-grained pixel level. Monakhova et al. (2022) took inspiration from physics-based noise formation in images, putting forward a temporal noise model that included a periodic and a time-variant row noise components to synthesize noises for starlight video denoising. Cao et al. (2023) considered all possible noise components, and established connection between each component and camera ISO through normalizing flow-based framework, solving parameter estimation problems in a complete model.

In spite of their ability to authentically portray aspects of noises inside camera sensors, these models are fundamentally different from EMCCD's. We intend to modify their methodology in order to construct best-suited noise model and calibration procedure for EMCCD denoising.

## 2.3 DEEP NEURAL NETWORK

U-Net (Ronneberger et al., 2015), has been a popular network structure in dense prediction tasks like semantic segmentation and image restoration. SwinIR (Liang et al., 2021) exploits both CNN and transformer, combining them into shifted window attention blocks to better aid image enhancement process. Restormer (Zamir et al., 2022) reduces complexity of transformer blocks through multi-head transposed attention and feed-forward network, achieving long-range pixel connections on large images, while Uformer (Wang et al., 2022) also increases its computational efficiency by non-overlapping window, depth-wise convolution, and skip connections, maintaining high performance in image enhancement tasks at the same time. Combining the evolved power of neural networks and data synthesis from noise modeling, we aim for a robust algorithm for EMCCD denoising.

## 3 NOISE MODEL

### 3.1 BASIC FORMATION

In figure 2, we present a concise yet precise noise model based on theoretical analysis and experimental results on iXon Ultra 897 EMCCD [4]. The reason for choosing this device is two-fold: on one hand, its manufacturer, Andor, part of Oxford Instruments, is well-regarded for its advanced EMCCD cameras that offer high performance in research areas, making their products widely adopted in both academia and industry. On the other hand, the stability of this device compared to others allows us to focus on challenging noise situations without being disturbed by components that can be avoided through hardware designs.

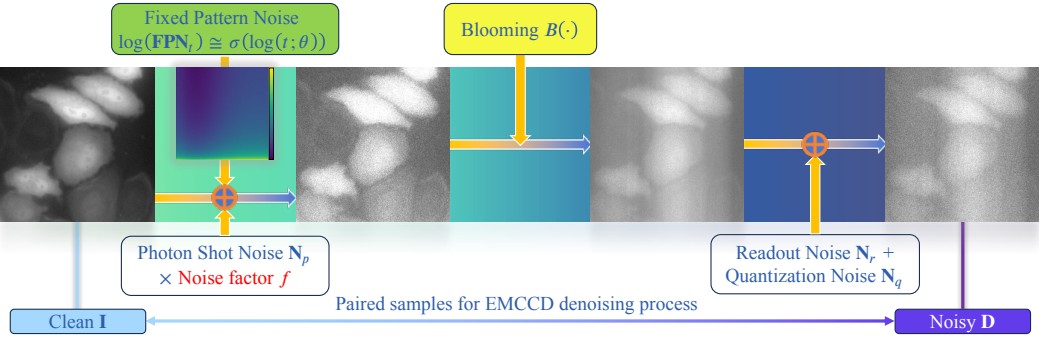

Figure 2: Outline of our EMCCD noise model, with 3 distinct advantages: inclusion of EMCCD's unique noise factor $f$; a novel logarithmic logistic regression correlation for fixed pattern noise and exposure; an unprecedented numerical model for blooming effect.

A mathematical description of noise components can be formulated as:

$$\mathbf{D} = B\left[K \times (\mathbf{I} + \mathbf{N}_P \times f) + \mathbf{FPN}\right] + \mathbf{N}_r + \mathbf{N}_q. \tag{1}$$

In equation 1[5], $\mathbf{I}$ is photon number that actually hit image sensor. $D$ is produced image, *i.e.*, digital number in pixels. $\mathbf{N}_P$ is Poisson shot noise, multiplied by noise factor $f \in \left[1.3, \sqrt{2}\right]$ to form a combination of shot noise and multiplicative noise as in Dussault & Hoess (2004). $K$ is system overall gain, a value that has absorbed quantum efficiency, EM gain, analog-to-digital conversion (ADC) factor. It can be viewed as a direct ratio between final digital number and incident photon count, if other noises are omitted. $\mathbf{FPN}$ stands for fixed pattern noise, a pixel-wise bias map of dark current non-uniformity. $B\left[\cdot\right]$ represents introduction of blooming artifacts, acting upon all components that induces photoelectrons. $\mathbf{N}_r$ is readout noise described by a zero-mean normal distribution, while $\mathbf{N}_q$ is quantization noise described by a uniform distribution.

As emphasized in figure 2, our model has 3 uniqueness: noise factor that only appears in EMCCD; precise $\log$-logistic regression for FPN; quantitatively handling blooming effect. Different from previous models that establish correlations between noise parameters and camera gain/ISO, we limited our model to the largest EM gain, $300\times$ in our device, based on a pragmatic viewpoint: it's the case that requires denoising the most, and the most challenging one as well. We instead model noise parameters in terms of exposure time. The reason for this choice is elaborated in the next subsection.

### 3.2 PARAMETER ESTIMATION

We utilize two categories of images for quantifying noise components: flat-field and biased frames. The former refers to a uniformly illuminated scene, while the latter is deprived of incident lights.

**Flat-field frames** are captured by aiming the camera at a uniformly lit white board surface with minimum roughness and unevenness, without any specular reflection. The lens on camera is set to

---

[4]Operating temperature were set to -60 °C for all following calibration and test image acquisitions.

[5]Bold symbols indicate 2D images, *e.g.*, $\mathbf{D} \in \mathbb{F}^{M \times N}$, with $M$ and $N$ being image's height and width

focus at infinity to further diminish possible inconsistency among expected value of pixels. These frames are used to calibrate system overall gain $K$ under each EM gain level.

Similar to ELD, a median-variance method can be deduced for EMCCD based on properties of Poisson distributions and multiplicative noise: $Var(\mathbf{D}) = Kf^2 \times Median(\mathbf{D}) + Var(\mathbf{N}_o)$, where $\mathbf{N}_o$ represents unrelated noise components combined. This implies that we can acquire $K$ by linear regression on variance and median of flat-field frames. Resulting slope would be the target $K$ times square of noise factor $f$.

Ryan et al. (2021) developed a gain-series approach based on EMCCD complete model uses maximum likelihood estimation to get more accurate $K$. For more details, please refer to the paper and experiments illustrated in our appendix. Estimated $K$ from two methods are not far apart, yet gain series approach's result has better linearity w.r.t. EM gains, which is why we adopt $K$ from gain series as parameters for data synthesis.

**Biased frames** are taken with EMCCD camera's lens cap on and camera being placed in a tightly sealed darkroom, to prevent any accidental photon response on the sensor. Contained in these almost completely dark images are systematic contribution from pixel non-uniformity and signal-independent fluctuations, *i.e.*, FPN and readout noise. To get rid of random variations, at any given exposure time, we took 1000 biased frames, pixel-wise averaging of which would produce FPN at that specific exposure. An example of FPN is displayed in figure 3. As can be seen from sub-figure

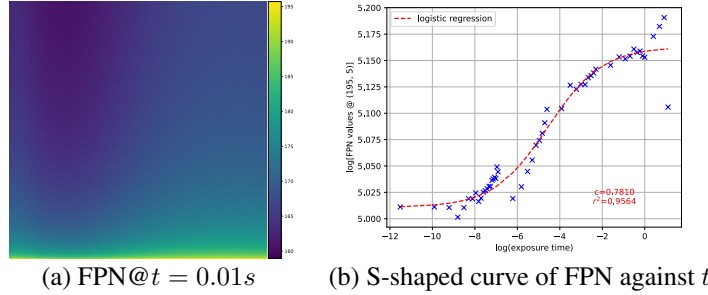

(a) FPN@$t = 0.01s$        (b) S-shaped curve of FPN against $t$

Figure 3: Characteristics of fixed pattern noise (FPN). (a) FPN when exposure is 0.01s; (b) Logarithmic logistic regression for pixel value of FPN at coordinate (195, 5) with respect to exposure.

(a) that, despite varying on a small scale, FPN does have a clear pattern with pixel values increasing from top to bottom, ranging from 160 to 195. In total, 52 exposure time were chosen lying between $10^{-5}$s to 3.0s. Even though there is no linear correlation between pixel value of FPN at any specific coordinate and exposure, we discovered that on logarithmic scale those values of a pixel and exposure form an S-shaped curve, namely, $\log(FPN_{ij}) \simeq \sigma(\log(exposure); \theta_{ij})$, with $FPN_{ij}$ being pixel value at coordinate $(i, j)$, $\sigma$ being the logistic curve, and $\theta_{ij}$ being coefficients for regression. In figure 3 (b), an example of this log-logistic regression is shown, demonstrating closeness between observed value and fit curve, having a coefficient of determinant, $r^2$, over 0.95. One possible explanation for this phenomenon is that, since non-uniformity bias of dark current response is reflected in FPN, such component could have a temporal accumulation, possibly from heat, and a saturation point, forming into an S-shaped curve on logarithmic level. After proper removal of outliers during regression, we managed to refine coefficients from a pixel-specific map to one group of values for entire FPN, *i.e.*, $\theta_{ij} \rightarrowtail \theta$. Using these coefficients and base FPN of $10^{-5}$s, we are able to achieve FPN predictions with average error per pixel below 1 digital number for exposure between 0.006s and 0.1s. This can be formulated as follows:

$$\mathbf{FPN}_t = \exp(\sigma(\log(t); \theta)), \tag{2}$$

with $t$ representing exposure.

As for readout noise, it is acquired by subtracting corresponding FPN from biased frames. Remaining signal variations follow closely to independent normal distributions at each pixel. In figure 4, plot (a) is a normal probability plot from one pixel calculated across 1000 samples under the same exposure. As empirical distribution (blue dots) overlaps with theoretical normal quantiles (red line), along with an $r^2$ over 0.99, it can be proved that readout noise follow pixel-wise normal distributions. However, those independent randomness does not share parameters. As is evident in plot (b)

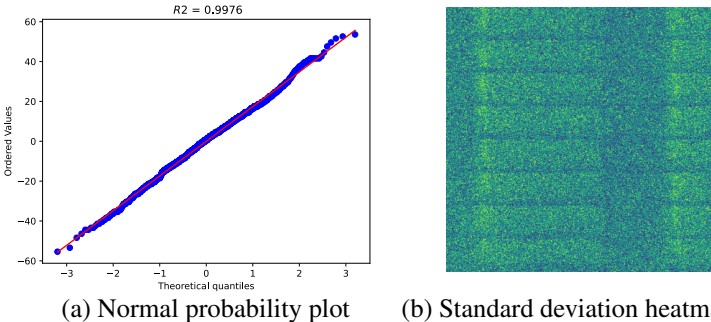

(a) Normal probability plot      (b) Standard deviation heatmap

Figure 4: Characteristics of readout noise at $t = 0.05s$. (a) Normal probability plot of readout noise at one pixel , with observed blue values perfectly aligned with a red line that represents standard normal distribution; (b) A heatmap (with enhanced contrast) of $std$ of readout noise at each pixel.

of figure 4, a standard deviation heatmap under one exposure shows spatial fluctuations. There is no strong connection between such std maps and exposure. Therefore, we store those $std$ maps and sample from them to generate normally distributed pixel-heterogeneous readout noises.

**Blooming** is another important artifact introduced by EMCCD's structural design independent of noise parameters. To verify and quantify those vertically extended light areas, we developed a new category of frame for blooming factor calibration as shown in figure 5. The images are also taken

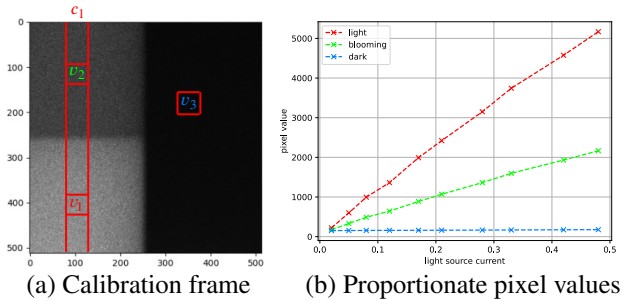

(a) Calibration frame      (b) Proportionate pixel values

Figure 5: Calibration of blooming factor. (a) Example of a calibration frame, which places a stable light source at the bottom left quadrant against a completely black background inside a sealed darkroom, leaving the top left quadrant to be mainly blooming pixels; (b) A plot of pixel values (y-axis) from 3 areas, light, blooming, dark, in the calibration frame under various light source intensities (x-axis), showing a proportionate correlation between blooming pixels and bright ones.

inside a darkroom, with a stable light source carefully placed to be at the bottom left quadrant of the frames against a black background, forming 3 distinct areas inside each frame: light, blooming, and dark. 1000 images were taken under each exposure time of 19 selected levels from 0.001s to 0.1s. Following calculations are all carried out on a pixel-wise averaged image from those 1000 images under each exposure in order to eliminate variations from the noises. Corresponding FPN of a given exposure is also subtracted from the averaged frame to reduce spatial non-uniformity. We sample mean values from sub-regions in 3 areas: $v_1$ from bright light source, $v_2$ from light source excited blooming, and $v_3$ from dark area. A plot of these 3 values under a series of varying light source intensities is shown in figure 5 (b), demonstrating a clear linearity for light induced blooming effect.

Based on this observation, we derived a simple estimation method to calculate a blooming factor $x$ solely from values obtainable in the calibration frame.

$$x = \frac{v_2 - v_3}{c_1 - v_2}, \tag{3}$$

where $c_1$ is a column-wise mean value of columns that contain bright area $v_1$ and blooming area $v_2$ as shown in figure 5. $x$ is defined as the ratio between blooming and bright pixels, which can be used in an inverted fashion to remove blooming in practice. Details of the derivation can be found in our appendix.

We demonstrate examples of blooming removal in figure 6, as evidence for validity of our hypothesis on the artifact.

(a) Blooming   (b) Removed   (c) Blooming   (d) Removed

Figure 6: Results of blooming removal. (a)&(b) Original image of an indoor macroscopic scene with obvious vertical streaks and its blooming removal result; (c)&(d) Microscopic scene with vertical light streaks and blooming removed from it.

### 3.3 SYNTHESIS

Inspired by Feng et al. (2024), we implement a similar preprocessing pipeline to deal with blooming and FPN outside of neural network, simplifying denoising training samples for the deep learning module. Specifically, for a given noisy input image $D$, our pipeline processes it as follows:

$$\mathbf{D}' = B^{-1}(\mathbf{D}) - \mathbf{FPN}_t = \left( \mathbf{D} - \frac{x_t}{1 + x_t} col\_mean(\mathbf{D}) \right) - \mathbf{FPN}_t. \tag{4}$$

$t$ refers to exposure time, $x_t$ being corresponding blooming factor calculated from equation 3, and $\mathbf{FPN}_t$ being corresponding fixed pattern noise described in equation 2. $col\_mean$ calculates column-wise averages. $\mathbf{D}'$ is the direct noisy input for our neural network. Combining equation 1 and 4, it is clear that remaining degradation includes shot noise, multiplicative noise, and readout noise. Therefore, given a clean image $\mathbf{S}$, we synthesize a noisy image $\hat{\mathbf{D}}'$ by:

$$\hat{\mathbf{D}}' = K\left(\mathbf{I} + \mathbf{N}_p \times f\right) + \mathbf{N}_r + \mathbf{N}_q; \tag{5}$$

$$\mathbf{I} = \frac{\mathbf{S}}{K} \; ; \; \mathbf{I} + \mathbf{N}_p \sim \mathcal{P}\left(\mathbf{I}\right) \; ; \; f \sim U\left[1.3, \sqrt{2}\right] \; ; \; \mathbf{N}_r \sim \mathcal{N}\left(0, \mathbf{std}_t^2\right) \; ; \; \mathbf{N}_q \sim U\left(-\frac{q}{2}, \frac{q}{2}\right). \tag{6}$$

$\mathcal{P}$ is Poisson distribution, $U$ uniform distribution, and $\mathcal{N}$ normal distribution. $K$ is system overall gain as in equation 1, while $\mathbf{std}_t$ is a pixel-wise standard deviation map from remaining readout noise in biased frames after subtracting FPN. $\mathbf{std}_t$ is randomly sampled from all the std maps we collected across multiple exposure time to increase robustness of learning process. $q$ is quantization step for the final digital number in each pixel.

## 4 EXPERIMENTS

### 4.1 TRAINING

We trained Uformer on synthetic data. Utilizing high definition raw images from SID, we randomly crop $512 \times 512$ samples from 231 images of size $4288 \times 2848$ to add noises. The GT clean images are originally in color format arranged in a Bayer-pattern, and are thus converted to single channel monochrome format before cropping. An example is shown in figure 7. Also included in the figure are two other noise modeling synthetic samples, one of which, shown in sub-figure (c) and denoted as ELD from Wei et al. (2021), is using a typical physics-based noise modeling for commercial CMOS-sensor cameras. It can be seen that, though recalibrated on our device, it emphasizes row-wise banding-pattern noise that is absent in EMCCD cameras. The other method is a theoretical complete EMCCD noise model by Ryan et al. (2021); Hirsch et al. (2013), shown in sub-figure (d). Main differences between our model and the theoretical one include:

1. preprocessing for blooming and FPN removal;
2. model of FPN and residual noise's dependency on exposure time;
3. instead of computationally expensive PGN model, we focus more on tractable yet crucial components for neural networks.

| (a) Clean | (b) Our noise | (c) ELD | (d) theoretical |

Figure 7: Synthesized noisy images. (a) A clean GT from SID; (b) Simulated noisy image using our proposed method, adding noise pattern while maintaining content and contrast; (c) Simulated noisy image using ELD, a typical noise model for ordinary camera sensors, introducing unnatural row noise that is not in EMCCD; (d) Simulated noisy image using theoretical EMCCD noise model from Ryan et al. (2021), slightly altering overall contrast of the image. All noise model are calibrated using our device.

For more details on ELD and theoretical EMCCD noise model, please refer to respective papers. We include other details of our training process in our appendix.

## 4.2 TESTSET AND FINETUNING

To the best of our knowledge, there is no real-world evaluation dataset for EMCCD. Thus, to evaluate trained networks quantitatively, we collected a first-ever EMCCD test dataset. Placing the camera at a stable position, and controlling its exposure and gain values through cable on a separated device, we attained long exposure clean images and short exposure noisy images with pixel-level alignment.

Exposure for clean GT images span a range from $0.5$s to $8.0$s depending on light intensity of the scene. Noisy input images can have exposure ranging from $1 \times 10^{-3}$s to 0.1s, creating ratios between GT and input's exposure ranging from 30 to 1600. GT images are all shot with EM gain disabled and averaged over 10 consecutive frames, while input images with EM gain set to 300 and no averaging. Total number of image pairs is 224, as one GT could correspond to multiple input images with various short exposure time. Example testset images can be found in the appendix.

Aside from quantitative measurements on a macroscopic testset, we also consider the practical usage of our method in microscopic observations. We further finetuned our model by synthesizing noise on a small dataset of 24 microscopic images including cytoplasm, endoplasmic reticulum, mitochondria, and nucleus. Each image are obtained by taking 100 consecutive $512 \times 512$ frames with 3s exposure, followed by an averaging process and median filter to suppress noise. Example images of the finetuning dataset and additional training details are included in our appendix.

## 5 RESULTS

### 5.1 QUANTITATIVE

We use peak signal-to-noise ratio (PSNR), structural similarity index measure (SSIM by Wang et al. (2004)), and learned perceptual image patch similarity (LPIPS by Zhang et al. (2018)) as our metrics to evaluate all results.

Table 1: Quantitative measurements of model performances. Best results are shown in red. The first row, Input, is direct evaluation on noisy inputs against clean GT, to serve as a frame of reference for other results. Our model outperforms all others by at least 2dB in PSNR.

| Methods | PSNR↑ | SSIM↑ | LPIPS↓ |
|---|---|---|---|
| Input | 40.73 | 0.9173 | 0.4007 |
| BM3D (Dabov et al., 2007) | 43.69 | 0.9662 | 0.1438 |
| ELD (Wei et al., 2021) | 43.90 | 0.9724 | 0.1187 |
| Theoretical (Ryan et al., 2021) | 46.44 | 0.9788 | 0.0987 |
| SRDTrans (Li et al., 2023) | 44.48 | 0.9677 | 0.2116 |
| Ours | 48.59 | 0.9854 | 0.0742 |

We compare our method against a traditional image processing denoising algorithm, BM3D from Dabov et al. (2007), and two aforementioned noise models for learning based supervised method, ELD and theoretical EMCCD noise model, with recalibrated noise parameters and same training process. In addition, a latest self-supervised fluorescence image denoising algorithm, SRD-Trans (Li et al., 2023), is compared as well. Measurements tested on paired macroscopic dataset are reported in table 1. From the table it can be seen that, although all algorithms introduce performance gain compared to direct input, our method has the best results among all, outperforming the second best by more than 2dB in PSNR.

We conducted ablation study on components of our noise model. In table 2, by excluding each part from our full model separately, it hinders overall performances to certain extents, providing evidence for optimality of our strategy.

Table 2: Ablation study on each component of our noise model. Removal of any one would result in performance loss, proving effectiveness of our ensemble design.

| Experiments | Multiplicative | Readout | FPN | Blooming | PSNR |
|:---:|:---:|:---:|:---:|:---:|:---:|
| I | ✗ | | | | 47.41 |
| II | | ✗ | | | 47.46 |
| III | | | ✗ | | 47.44 |
| IV | | | | ✗ | 46.52 |
| IV | ✓ | ✓ | ✓ | ✓ | 48.59 |

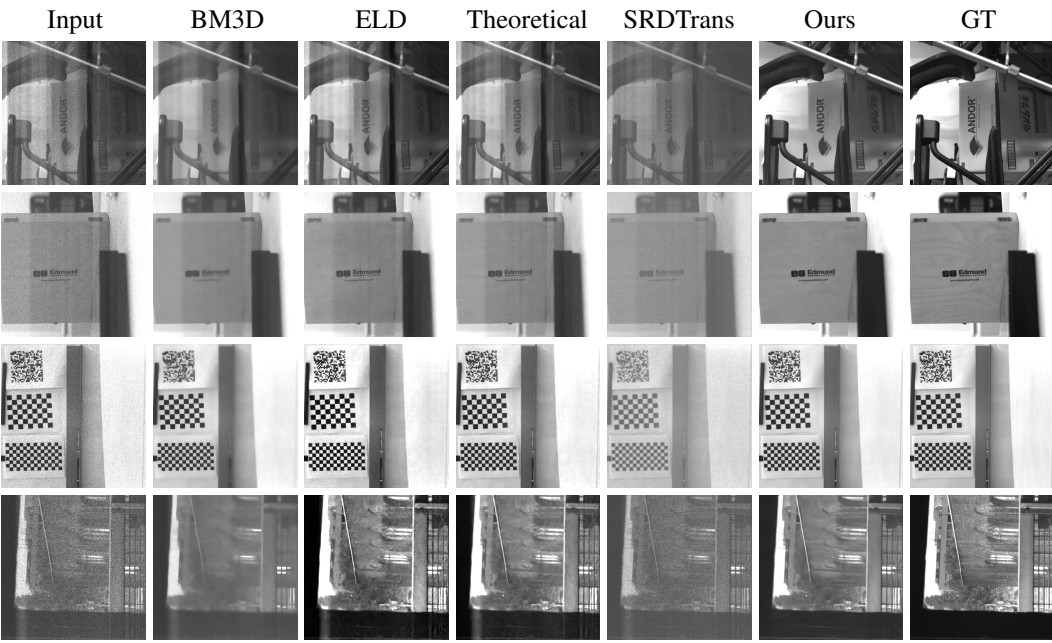

Figure 8: Qualitative results of samples from our macroscopic testset. Each row represents one sample, while each column is input/result from different methods annotated at the top of the figure.

## 5.2 QUALITATIVE

We exhibit qualitative test results of our method and other compared models in figure 8. It can be seen from the images that not only does our model handle blooming better than others, but overall remaining fluctuations in our results are minimum as well, producing closest appearances to GTs.

In figure 9, we show results on fluorescence microscopic images, where clean GTs are not available. BM3D, ELD, theoretical model, and SRDTrans have conspicuous flaws of oversmoothing, grid-

Input  BM3D  ELD  Theoretical  SRDTrans  Ours  Finetuned

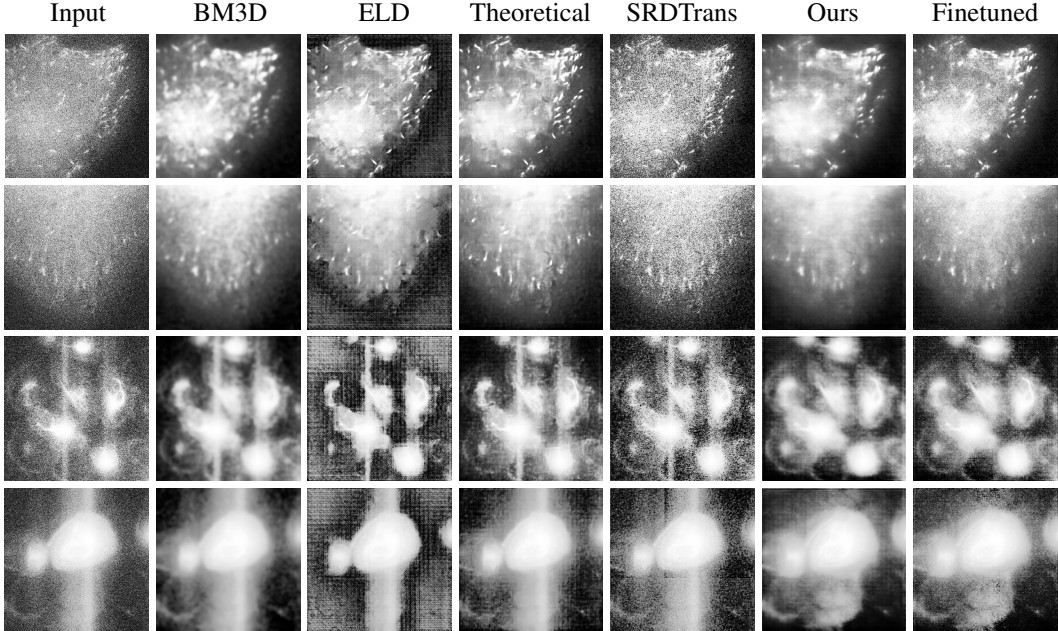

Figure 9: Qualitative results of samples from fluorescence microscopic testset. Each row represents one sample, while each column is input/result from different methods annotated at the top.

pattern artifacts, inability to handle blooming, and strong remaining noise. Our model that is trained on macroscopic synthesized data produces smoothed results, while after finetuning on microscopic images, the model could reach a balance between denoising and detail preserving, generating sharp edges around high contrast areas and maintaining certain level of noises for smooth transition at low contrast areas to avoid artifacts.

Zoomed-in versions of qualitative results can be found in our appendix, providing more detailed visualizations.

## 5.3 DISCUSSION

Despite achieving visually pleasant output, our trained network still faces inevitable overly-smoothed and residual-noise problems in some test samples, as in all other denoising tasks. Strong noise intensities entail large randomness making it hard to acquire accurate noise modeling, while neural networks demand samples that are representative of authentic noise distributions. The solution we offered is a gateway to access pioneering statistical learning for EMCCD denoising. Possible improvements on this path include: self-supervised learning (Yang et al., 2024) or few-shot-learning (Jin et al., 2023) for noise modeling to relieve the burden of calibration data collection; building larger microscopic dataset like Biswas & Barma (2020) that enables high capacity networks for more accurate noise removal; incorporation of generative diffusion models (Ho et al., 2020) to fill in missing information caused by strong noises.

## 6 CONCLUSION

In this paper, we proposed a novel physics-based noise model for EMCCD cameras. A complete systematic noise parameter calibration process is established based on both theoretical analysis and empirical experiment results, including unique logarithmic logistic regression for fixed-pattern noise and effective blooming removal technique. Synthesizing noisy images from a real-world low-light raw image dataset, we trained a state-of-the-art deep learning model. A world's first macroscopic paired testset for EMCCD is collected, experiments on which demonstrated superiority of our model against preexisting methods in image processing, noise modeling in computer vision, and theoretical analysis for EMCCD noise. Our model can be further finetuned on microscopic dataset, elevating image quality of denoising output for EMCCD shot fluorescence microscopic images.

ACKNOWLEDGMENTS

This research was supported in part by JST-Mirai Program JPMJMI23G1, JST CREST Grand Number JPMJCR15N3, JSPS KAKENHI Grant Numbers 24KK0209, 24K22318, 22H00529, 18H05410 and 22K04891.

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

# A CALIBRATION DETAILS

## A.1 FLAT-FIELD FRAMES

In ELD (Wei et al., 2021), a mean-variance method is adopted for estimating $K$, leveraging following correlation based on property of Poisson distributions:

$$
\begin{aligned}
Var(\mathbf{D}) &= K^2 Var(\mathbf{I} + \mathbf{N}_p) + Var(\mathbf{N}_o) \\
&= K^2 \mathbf{I} + Var(\mathbf{N}_o) \\
&= K(K\mathbf{I}) + Var(\mathbf{N}_o) \\
&= K \times Mean(\mathbf{D}) + Var(\mathbf{N}_o).
\end{aligned}
\tag{7}
$$

$N_o$ represents all noise components other than shot noise combined. The proportional relation between variance and mean value of flat-field frames demonstrates that a linear regression could yield $K$. In EMCCD's case, we substitute $N_p$ in equation 7 with multiplicative noise $\mathbf{N}_p \times f$, updating the final linear relationship to be $Var(\mathbf{D}) = Kf^2 \times Mean(\mathbf{D}) + Var(\mathbf{N}_o)$, *i.e.*, the slope coefficient of regression is $Kf^2 = 2K$. In practice, flat-field frames are limited to a $256 \times 256$ crop at the center to avoid being affected by lens vignetting. Median, instead of mean value, is utilized to calculate independent variable in linear regression so as to increase robustness of the results.

Shown in figure 10 is a comparison between estimated system overall gain $K$ by mean-variance method and the gain-series approach from Ryan et al. (2021) on flat-field frames. As mentioned

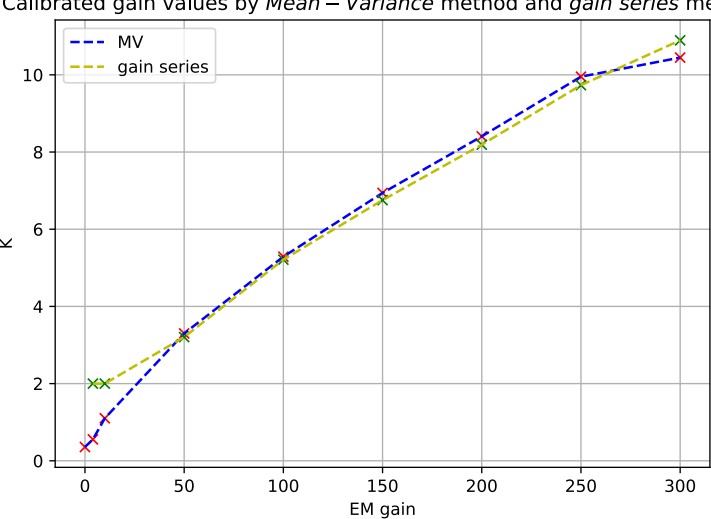

Figure 10: System overall gain $K$ estimated by MV and gain series method.

in the main text, the estimates of $K$ from both methods are fairly close; however, the gain series approach exhibits better linearity with respect to the EM gains. Therefore, we adopt $K$ from the gain series for data synthesis.

## A.2 BLOOMING FACTOR

We started with a hypothesis that the essence of blooming is adding an amount that is proportional to column mean value to all pixels in that column. Borrowing notations declared in the main text and figure 5, for blooming pixels we can have:

$$v_2 = b + cx, \tag{8}$$

with $b$ representing dark background bias [6], $c$ being column mean without blooming, and $x$ as the blooming factor. Nevertheless, real column mean value without blooming is not directly obtainable from the calibration image, as the observable column mean from the image, denoted as $c_1$, is also increased by the blooming factor $x$, meaning that $c_1 = (1 + x)c$. Therefore, for bright and dark pixels:

$$v_2 = b + \frac{c_1}{1 + x}x, \tag{9}$$

$$v_3 = b + bx. \tag{10}$$

Combining the above two equations, we can have

$$x = \frac{v_2 - v_3}{c_1 - v_2}, \tag{11}$$

calculating the blooming factor $x$ solely from values obtainable in the calibration frame.

In practice, we empirically use a window size of 32 pixels to select sub-regions for $v_2$, $v_3$, and $c_1$ calculation, all of which are determined by a sliding-window search for sub-regions with minimum standard deviation among their pixels, in order to avoid any possible spatial heterogeneity. Result estimations of the factor are shown in figure 11, matching our experiment results that images taken with shorter exposure have stronger blooming effect.

Evidence of the validity of estimating blooming factor and remove such effect is included in our main text and in figure 6.

---

[6]Despite high photon absorption rate on dark surfaces in our dark room, there is still chance for photons to bounce off and fall into the image sensor, creating responses above dark current bias.

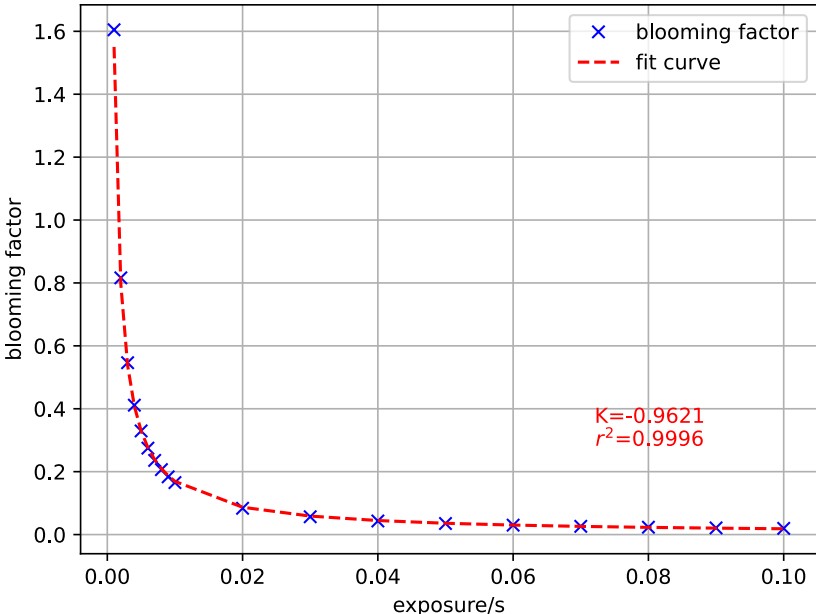

Figure 11: Estimated blooming factors under each exposure time. Correlations between the two are close to a hyperbolic curve. A log-linear regression yields a conforming result with exponential coefficient being -0.9621 (approximating -1 for reciprocal function) and an $r^2$ value of 0.9996.

## B  EXPERIMENT DETAILS

### B.1  TRAINING PROCESS

Training process for all experiments have the same hyperparameters, using AdamW optimizer (Loshchilov & Hutter, 2017) with initial learning rate as $2 \times 10^{-4}$ and 3 epochs for warmup training. Batch size is set to 4, training patch size to 512, number of total epochs to 1000. The rest parameters for Uformer are maintained as their default values. Trained with 2 RTX-3090 GPUs, each training process costs approximately 25 hours.

### B.2  TESTSET AND FINETUNING

Example pictures of our testset are shown in figure 12. Examples of clean microscopic images used for finetuning process are displayed in figure 13. The finetining proceeds for another 1000 epochs on synthetic noisy-clean image pairs from this dataset, continuing from previously trained checkpoints with all other hyperparameters unchanged and taking roughly 6 hours on the same hardware as before.

## C  ADDITIONAL EXPERIMENT RESULTS

We include here enlarged qualitative results from figure 8 and figure 9 respectively in the following. Figure 14 shows zoomed-in areas on macroscopic testset results, displaying the clearest edges around characters and shapes in our results, and at the same time with minimum remaining noise, least of noticeable artifacts.

In figure 15, apart from eliminating blooming effects, our results can preserve natural transitions between high and low contrast areas, balancing noise remnant and sharpness preservation while not introducing unwanted artifacts.

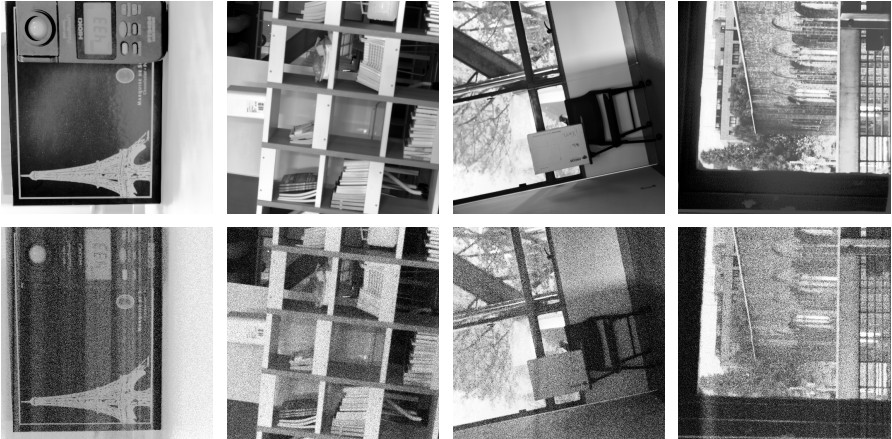

Figure 12: Sample images from our collected testset. The first row is ground truth clean images, acquired by long exposure and averaging across multiple frames. The second row is input noisy images, acquired by short exposure at the same position of corresponding GT.

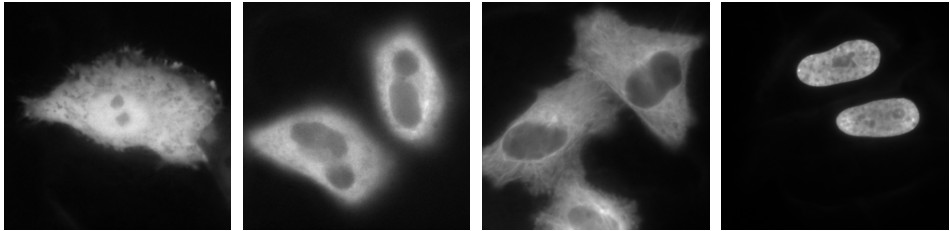

Figure 13: Examples of clean microscopic images, imaging cytoplasm, endoplasmic reticulum, mitochondria, and nucleus.

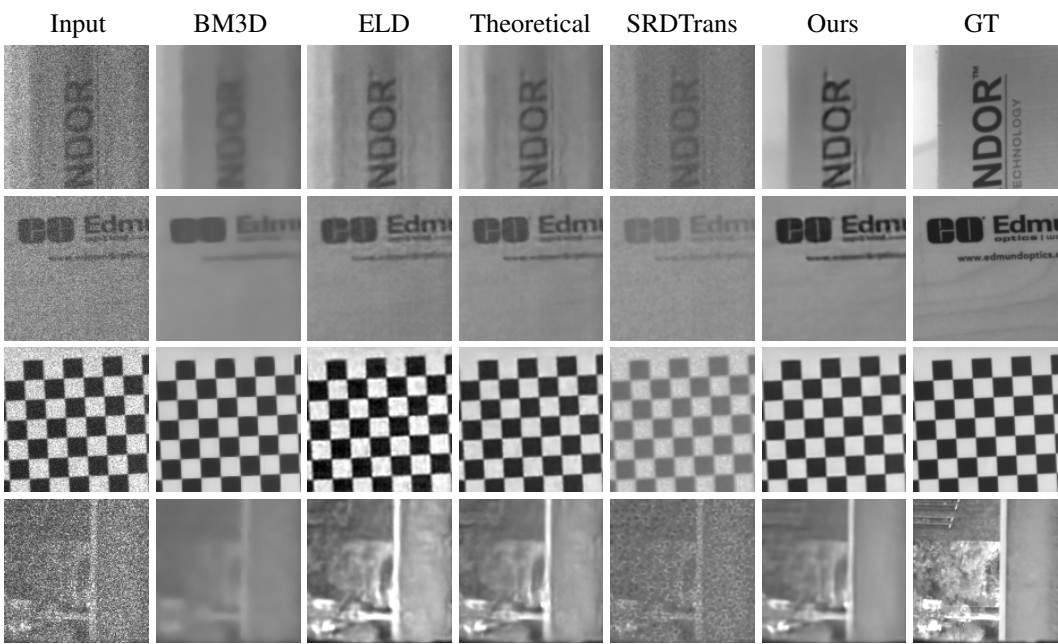

Figure 14: Enlarged qualitative results of figure 8 from our macroscopic testset. Notice the clarity of alphabets' edges produced by our results compared to others.

| Input | BM3D | ELD | Theoretical | SRDTrans | Ours | Finetuned |
|-------|------|-----|-------------|----------|------|-----------|

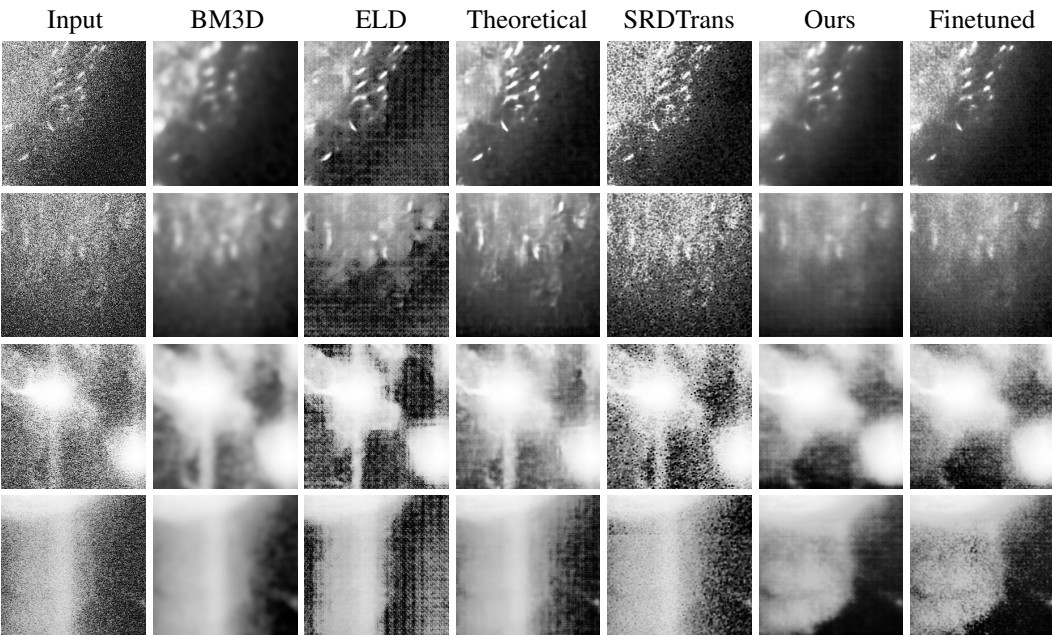

Figure 15: Enlarged qualitative results of figure 9 from fluorescence microscopic testset. Notice more natural transitions from high contrast areas to low ones and less artifact in our results.

