# OpenReview forum: "Revolutionizing EMCCD Denoising through a Novel Physics-Based Learning Framework for Noise Modeling"
_ICLR.cc/2025/Conference — ICLR 2025 Poster_

### Official Review · Reviewer_V6zx · 2024-11-02

**Soundness:** 3
**Presentation:** 4
**Contribution:** 3
**Rating:** 8
**Confidence:** 5

**Summary:**

This paper presents a novel approach to denoising images captured by electron-multiplying charge-coupled devices (EMCCDs) by introducing a physics-based noise model and a calibration procedure tailored for EMCCD-specific noise characteristics. The proposed method synthesizes authentic training data for a deep learning framework, enhancing denoising performance in fluorescence microscopy and achieving state-of-the-art results compared to existing methods. Additionally, they establish a comprehensive pipeline that connects noise parameter calibration with advanced neural network training strategies. This work paves the way for improved image quality in sensitive imaging applications across various scientific fields.

**Strengths:**

1. The introduction and the method of this paper are clear and easy to understand. Even readers who may not be familiar with EMCCD can grasp the motivation behind the noise model.

2. The novelty of this work is commendable. While many key designs are inspired by existing research, they incorporate unique adjustments specific to the characteristics of EMCCD sensors. The analysis of FPN, blooming effects, and readout noise heatmaps is particularly impressive.

3. The experiments presented in this paper are excellent, and I believe they will significantly contribute to sensitive imaging applications across various scientific fields.

**Weaknesses:**

1. In Eq. (5), D' includes $N_r$ and $N_q$; however, this seems unreasonable from a formulaic perspective. I suggest explaining why $B^{-1}$ doesn't affect $N_r$ and $N_q$. For instance, it might be beneficial to analyze the expected interactions between these two components.

2. Figure 3(b) appears to exhibit some abrupt transition points (e.g., log(time) = -7, -4), and the explanation provided in L252-255 seems insufficient to cover this phenomenon. Please confirm the reproducibility of these data and clarify why an S-shaped curve is used instead of multiple piecewise functions. If these transition points are related to circuit switching, a piecewise function fitting, similar to what has been reported in PMN, should be employed.

**Questions:**

### Original Question
The relationship between the ablation study and the proposed method in this paper is unclear.

As it stands, I find it difficult to directly correlate the FPNt, blooming effect, and readout noise heatmap with the ablation learning presented. Additionally, the current preprocessing appears to resemble contributions from PMN rather than from this work.

I suggest clarifying the incremental contributions of the proposed method in the experiments to emphasize the original contributions of this paper.

### After Rebuttal
The authors have addressed my concerns.

I found the authors’ response to reviewer e1uJ very well-written. This work stands out because it takes a practical, problem-specific approach, using appropriate innovations to solve a real-world task. Looking at recent noise modeling research, I consider LLD [1] and PNNP [2] to be practical, while LRD [3] seems less so. LRD faces challenges with the data dependency problem, whether **paired data or noise models come first**, and the instability of the GAN-based training strategy. As a result, GAN-based noise modeling methods like LRD, CA-GAN, and Starlight often overfit the training data.
For example, LRD, which includes a **Fournier Transformer Discriminator**, leaves visible row patterns in *Scene-07_IMG-0010* of the ELD dataset that are more noticeable than those in the ELD baseline, even though LRD achieves higher PSNR and SSIM scores. For this reason, I believe reviewer e1uJ’s initial rejection of this work was not well-founded.

In conclusion, I acknowledge the contributions of this paper and am inclined to keep my current rating.

**Reference**
[1] Y. Cao, M. Liu, S. Liu, X. Wang, L. Lei, and W. Zuo, ‘Physics-Guided ISO-Dependent Sensor Noise Modeling for Extreme Low-Light Photography’, in Proceedings of the IEEE/CVF Conference on Computer Vision and Pattern Recognition (CVPR), 2023, pp. 5744–5753.
[2] H. Feng, L. Wang, Y. Huang, Y. Wang, L. Zhu, and H. Huang, ‘Physics-guided Noise Neural Proxy for Practical Low-light Raw Image Denoising’, arXiv preprint arXiv:2310. 09126, 2023.
[3] F. Zhang et al., ‘Towards General Low-Light Raw Noise Synthesis and Modeling’, in Proceedings of the IEEE/CVF International Conference on Computer Vision (ICCV), 2023, pp. 10820–10830.

---

> ### Author Response · Authors · 2024-12-02
> **Response to Reviewer V6zx**
>
> We thank Reviewer *V6zx* for the constructive comments and address the concerns as follows:
>
> 1. Theoretically speaking[1](https://andor.oxinst.com/learning/view/article/ccd-blooming-and-anti-blooming):
> > Blooming occurs when the charge in a pixel exceeds the saturation level and the charge starts to fill adjacent pixels.
>
> In traditional CCD cameras, this phenomenon occurs prior to read-out circuits, which precludes $N_r$ and $N_q$ that happen relatively downstream in image formation.
> However, the exact definition we included in our formula (4) & (5) is determined by the process of how we calibrate blooming factor.
> As shown in Fig.5, calibration frames for this factor are the end results of the entire image acquisition, which means we are unable to verify intermediate results to see when each components would be added into the frame.
> Having said that, as Fig.5 (b) has already shown relative good linearity of blooming effect, we strip blooming effect from all components from actually acquired image $D$, as indicated in equation (4).
> In equation (5), it is the synthetic image for network training.
> As long as blooming removal in equation (4) is in accordance with its calibration from equation (3), we believe it is a coherent reasoning, which is corroborated by results of effective blooming removal in Fig.6.
>
>
> 2. FPN Calibration:
> We acquired FPN calibration frames multiple times, all of which can be fitted in S-shaped curve, with an accuracy of less than 1 digital number margin of error averaged over all pixels for exposure between 0.006s and 0.1s. However, there do exist different transition smoothness between different acquisitions, which cannot be fully explained with our current observations, and we will continue to explore its reasons. On the other hand, piecewise linear regression for FPN representations produce slightly worse results than our S-shaped regression version: PSNR of piecewise regression yields 47.98dB vs 48.59dB in our paper.  Even though piecewise function can have better fitting performance, it is possible that such representation is more prone to overfitting of FPN dark frames than using a more general and smooth S-shaped regression.
>
> 3. Ablation Study:
> We genuinely appreciate the suggestion on our ablation study. We conducted finer-grained ablation studies to evaluate contributions from preprocessing components like FPN and blooming removal. The table below summarizes the results:
>
> | Experiment | Multiplicative | Readout | Preprocessing | FPN | Blooming |  PSNR |
> |:----------:|:--------------:|:-------:|:-------------:|:---:|:--------:|:-----:|
> |      Ⅰ     |        ✖       |         |               |     |          | 47.41 |
> |      Ⅱ     |                |    ✖    |               |     |          | 47.46 |
> |      Ⅲ     |                |         |       ✖       |     |          | 44.96 |
> |      Ⅳ     |                |         |               |  ✖  |          | 47.44 |
> |      Ⅴ     |                |         |               |     |     ✖    | 46.52 |
> |      Ⅵ     |        ✔       |    ✔    |       ✔       |  ✔  |     ✔    | 48.59 |
>
> These results indicate that our preprocessing is distinct from PMN. Notably, blooming removal contributes significantly to performance gains, while the combination of FPN and blooming removal provides a substantial boost.

---

### Official Review · Reviewer_M8ik · 2024-11-03

**Soundness:** 2
**Presentation:** 2
**Contribution:** 2
**Rating:** 5
**Confidence:** 3

**Summary:**

This paper focuses on EMCCD noisy data and denoising. The authors propose a physics-based noise model specifically for EMCCD cameras, which generates synthetic noisy images based on both the camera's properties and EMCCD-specific noise characteristics. This method makes the data clorser to real-life scenarios. They then train a deep learning model, Uformer, on these noisy image pairs for denoising. The Uformer model achieves better denoising results comparing to other methods.

**Strengths:**

1. Proposed the first dataset specific to EMCCD.
2. Provides a detailed and clear explanation of the noise model, including settings and parameter estimation.
3. The experiments compare the proposed method to other state-of-the-art methods.

**Weaknesses:**

1. For important equations, such as Eq. (1) and (5), the dimensions of each parameter are not provided, especially for N_p, f, and I.
2. The denoising model, Uformer, should be discussed more thoroughly, with additional details explaining its design, such as the key differences compared to the Uformer model from Wang et al., 2024.
3. The total number of image pairs is 224, which is relatively small, and the use of only 24 images for fine-tuning could lead to overfitting.

**Questions:**

1. Could you provide the dimensional details of the variables in the key equations listed in the paper? It would help in understanding if you state that 'X' represents the inner product in Eq. (1).
2. It seems that adding N_r and N_q makes the image blurry. Could you visualize both N_r and N_q?
3. I feel that the proposed noise addition might be similar to the negative binomial low-photon noise. Could you explain the key differences between them?
4. In line 076, could you elaborate on the differences between the EMCCD and other models, if possible?
5. Could you provide a big-map plot or additional explanation of your Uformer model? What is the novel design aspect of this denoising model, and how does it differ from Wang's model?
6. Did you use any method to measure whether the results indicate overfitting? Will using data augmentation techniques to generate more data improve the model's accuracy? Perhaps training the model on simulated data and testing it on the original true data could be a way to assess the quality of the simulated data.

---

> ### Author Response · Authors · 2024-12-02
> **Response to Reviewer M8ik**
>
> We thank Reviewer *M8ik* for the valuable feedback and offer the following clarifications and responses:
>
> 1. Regarding the notation in Equation (1), most of the variables are 2D images, *i.e.* $D, I, N_p, \text{FPN}, N_r, N_q \in \mathbb{F}^{M \times N}$ (with $M$ and $N$ being height and width of the image), while $K$ and $f$ are scalars. The transformation $B[\cdot]$ maps one image to another. All $\times$ symbols represent element-wise multiplication between a matrix and a scalar. We appreciate the suggestion to make the notation more informative. In the final version, vectors and matrices will be bolded, and their dimensionalities will be explicitly defined to prevent confusion.
>
> 2. From Information Theory perspective, noise patterns such as $N_r$ and $N_q$ inevitably cause content loss, manifesting as visual distortions like blurriness in spatial domains. The authenticity of these noise components can be verified by comparing our synthesized noisy images with real noisy images and through the performance gains achieved by our pipeline over existing methods.
>
> 3. In the context of low-photon count imaging, the negative binomial distribution is a method to explain variabilities larger than mean values of the signals, since Poisson statistics dictates equality between mean and variance. However, Poisson distribution is only for photon shot noise, which is the signal-dependent components in our paper. Low-photon count imaging can suppress read-out noise from imaging circuits, but not able to eliminate all such noises. Thus, our additive noise components are not simply for explanation of long-tail traffic of photon incidences, but rather a description of possible independent noise patterns from specific device physics.
>
> 4. Key differences between EMCCD and other CMOS noise modeling are reflected in Fig.2 of our paper as uniqueness of our model, and also summarized here as follows:
>
> 	• EMCCD noise parameters are strongly influenced by exposure time, as accumulated heat affects the sensors, unlike CMOS models that primarily depend on system gain or ISO.
> 	• EMCCDs exhibit blooming effects, which are largely absent in CMOS models.
> 	• EMCCDs benefit from simpler read-out circuits, whereas CMOS models involve complex noise correlations and banding patterns.
>
> 5. For fair comparisons, we retained the original Uformer architecture, only modifying its input/output dimensions to handle single-channel grayscale EMCCD images. A tailored network to further improve denoising is part of our future work.
>
> 6. We collected real world data shot on an EMCCD device as shown in Fig. 8 in our appendix. All validation results (Tab.1 & 2), as well as qualitative demonstrations (Fig. 8, 9, 14, 15), are produced by training a network on synthetic data and testing it on real world data, so that risk of overfitting to synthesized data can be minimized in our conclusions.

---

### Official Review · Reviewer_e1uJ · 2024-11-03

**Soundness:** 3
**Presentation:** 3
**Contribution:** 2
**Rating:** 3
**Confidence:** 5

**Summary:**

This paper proposes a physics-based noise model for EMCCD cameras. The statistical model includes some typical noise components for EMCCD sensors, and a calibration method is proposed for adaptation this noise model on each sensor. Through careful noise modeling and calibration, the authors synthesize realistic EMCCD noise data for training, and effectively improve the learning of deep denoisiers in both macroscopic testset and microscopic testset.

**Strengths:**

- The paper introduces the first EMCCD denoising method utilizing physics-based noise modeling method.
- The overall writing of this paper is good and easy to follow.

**Weaknesses:**

- This paper proposes the first noise modeling method for EMCCD sensors, and there are indeed some new adaptations on this sensor type. However, the main idea borrows many contributions from the similar task of CMOS noise modeling, and seems to be a EMCCD-version of ELD [1]. Specifically, the entire pipeline, i.e., physics-based noise modeling ->  calibration -> synthesis -> denoise pipeline is the same with ELD. The noise components and calibration process are also similar with ELD. In addition, the modeling of FPN and pre-processing operation comes from PMN [2] .
- For Fig. 7, why ELD presents banding patterns, even after calibration using the target device? ELD calibrates row noise using bias frames, and the variance for row noise would be close to zero on sensors without obvious banding patterns if correctly calibrated. I wonder why ELD still causes such row patterns on EMCCD sensors.
- There should be more comparisons with sota methods, for both noise modeling and self-supervised denoising methods. For example, [3] proposes a general noise modeling method which uses poisson sampling for signal-dependent noise and GAN for signal-independent noise. I think [3] can also handle EMCCD sensors. Stronger baselines for self-supervised methods are also recommended to compare [4].
- I concern that it is not rigorous to use SID clean images to synthesize noisy pairs for training. Different from EMCCD sensors, SID dataset uses Sony cameras with CMOS sensors. Each sensor type has its own unique recipe for generating RAW data; even using clean images from one type of CMOS sensor to generate synthetic noisy pairs and then testing on real data from a different CMOS sensor can lead to negative effects, not to mention EMCCD data. Therefore, I believe that SID clean data is not a suitable choice for this application.
- Section 2.3 is not necessary since no deep denoiser architecture is proposed.


[1] Physics-based Noise Modeling for Extreme Low-light Photography. TPAMI 2021
[2] Learnability Enhancement for Low-Light Raw Image Denoising- A Data Perspective. TPAMI 2023
[3] Towards General Low-Light Raw Noise Synthesis and Modeling. ICCV 2023
[4] Exploring Efficient Asymmetric Blind-Spots for Self-Supervised Denoising in Real-World Scenarios. CVPR 2024

**Questions:**

Why ELD presents banding patterns in Fig. 7?

---

> ### Author Response · Authors · 2024-12-02
> **Response to Reviewer e1uJ**
>
> We thank Reviewer *e1uJ* for the insightful comments and appreciate the opportunity to address the questions raised.
>
> 1. While there are similarities between our method and typical CMOS noise modeling, our objective was not to entirely redefine the physics-based noise modeling framework. Notable works, such as those by LLD[1] and PNNP[2], have sought to enhance the performance of existing models within the same framework. In contrast, the primary goal of our research is to revolutionize the denoising of EMCCD sensors by introducing novel noise modeling techniques alongside comprehensive experiments and analyses. This approach aims to fully leverage the potential of EMCCD technology, which has yet to be thoroughly explored.
>
> 2. The banding patterns observed in ELD-simulated noisy samples arise because ELD treats channel biases as scalars. As shown in Fig. 3, FPN exhibits distinct spatial variations. Using a single scalar to compute bias values results in residual deviations from zero—for example, the pixel values in the top and bottom rows of a frame remain farther from zero even after DC bias correction. These residual deviations are the root cause of the banding patterns, as they are encapsulated in row mean values, leading to elevated $\sigma_{\text{row}}$ levels.
>
> 3. We conducted a comparison experiment to LRD, which yields a PSNR of 45.79dB, between theoretical noise analysis based synthesis (Ryan et al. 2021) and SRDTrans (Li et al. 2023), all of which worse than our results. Possible reasons for LRD's shortcomings include differences between CMOS and CCD sensors, its inability to handle spatially correlated noise patterns (such as combined FPN and blooming effects), and its reliance on extensive metadata to map noise to camera parameters. By contrast, EMCCD images offer limited metadata, making LRD less effective in this context. Our uniquely designed EMCCD noise model addresses these challenges effectively.
>
> 4. To minimize differences between CMOS and EMCCD sensors, we converted SID clean images into single-channel images before generating synthetic noisy data. We quantitatively evaluated all methods on a macroscopic scene captured with a genuine EMCCD device. Since the SID dataset contains similar content, our experiments sufficiently demonstrate the effectiveness of our pipeline as a proof of concept. Furthermore, our validation on a microscopic dataset underscores the pragmatic and ready-to-use nature of our method.
>
>
> _References_
>
> [1] Y. Cao, M. Liu, S. Liu, X. Wang, L. Lei, and W. Zuo, ‘Physics-Guided ISO-Dependent Sensor Noise Modeling for Extreme Low-Light Photography’, in Proceedings of the IEEE/CVF Conference on Computer Vision and Pattern Recognition (CVPR), 2023, pp. 5744–5753.
>
> [2] H. Feng, L. Wang, Y. Huang, Y. Wang, L. Zhu, and H. Huang, ‘Physics-guided Noise Neural Proxy for Practical Low-light Raw Image Denoising’, arXiv preprint arXiv:2310. 09126, 2023.
>
> [3] F. Zhang et al., ‘Towards General Low-Light Raw Noise Synthesis and Modeling’, in Proceedings of the IEEE/CVF International Conference on Computer Vision (ICCV), 2023, pp. 10820–10830.

---

### Meta-Review · Area_Chair_TiJC · 2024-12-20

**Metareview:**

The reviewers agree about the novelty of the approach and its application. Reviewer e1uJ had some concerns regarding the simulation results and the quality, which were addressed by the authors. Reviewer V6zx also confirms the justification by authors rebuttal to other reviewers.
AC has reviewed the comments from reviewer e1uJ and the authors response. Given that most of the questions are answered by the authors and that the reviewer did not change their original score, AC decided to remove their score from the evaluation.
Therefore the paper is recommended for acceptance.

**Additional Comments On Reviewer Discussion:**

Reviewer V6zx made a comment on other reviewer's criticism, but there were no response after that.

---

### Decision · Program_Chairs · 2025-01-22

Accept (Poster)